# Indigo Naturalis Ameliorates Dextran Sulfate Sodium-Induced Colitis in Mice by Modulating the Intestinal Microbiota Community

**DOI:** 10.3390/molecules24224086

**Published:** 2019-11-12

**Authors:** Yan-Ni Liang, Jin-Gao Yu, Dong-Bo Zhang, Zhen Zhang, Lang-Lang Ren, Lu-Han Li, Zheng Wang, Zhi-Shu Tang

**Affiliations:** Shaanxi Collaborative Innovation Center of Chinese Medicinal Resources Industrialization, State Key Laboratory of Research & Development of Characteristic Qin Medicine Resources (Cultivation), Shaanxi Innovative Drug Research Center, Shaanxi University of Chinese Medicine, Xian Yang 712083, China; 1501009@sntcm.edu.cn (Y.-N.L.); jingao_yu@sina.cn (J.-G.Y.); symensu@163.com (D.-B.Z.); zhzh626@outlook.com (Z.Z.); renlanglang2580@126.com (L.-L.R.); liluhan1127@163.com (L.-H.L.)

**Keywords:** indigo naturalis, colitis, DSS, gut microbiota, 16S rDNA

## Abstract

Indigo naturalis (IN) is a traditional Chinese medicine, named Qing-Dai, which is extracted from indigo plants and has been used to treat patients with inflammatory bowel disease (IBD) in China and Japan. Though there are notable effects of IN on colitis, the mechanisms remain elusive. Regarding the significance of alterations of intestinal flora related to IBD and the poor water solubility of the blue IN powder, we predicted that the protective action of IN on colitis may occur through modifying gut microbiota. To investigate the relationships of IN, colitis, and gut microbiomes, a dextran sulfate sodium (DSS)-induced mice colitis model was tested to explore the protective effects of IN on macroscopic colitis symptoms, the histopathological structure, inflammation cytokines, and gut microbiota, and their potential functions. Sulfasalazine (SASP) was used as the positive control. Firstly, because it was a mixture, the main chemical compositions of indigo and indirubin in IN were detected by ultra-performance liquid chromatography (UPLC). The clinical activity score (CAS), hematoxylin and eosin (H&E) staining results, and enzyme-linked immunosorbent assay (ELISA) results in this study showed that IN greatly improved the health conditions of the tested colitis mice, ameliorated the histopathological structure of the colon tissue, down-regulated pro-inflammatory cytokines, and up-regulated anti-inflammatory cytokines. The results of 16S rDNA sequences analysis with the Illumina MiSeq platform showed that IN could modulate the balance of gut microbiota, especially by down-regulating the relative quantity of *Turicibacter* and up-regulating the relative quantity of *Peptococcus*. The therapeutic effect of IN may be closely related to the anaerobic gram-positive bacteria of *Turicibacter* and *Peptococcus.* The inferred metagenomes from 16S data using PICRUSt demonstrated that decreased metabolic genes, such as through biosynthesis of siderophore group nonribosomal peptides, non-homologous end-joining, and glycosphingolipid biosynthesis of lacto and neolacto series, may maintain microbiota homeostasis during inflammation from IN treatment in DSS-induced colitis.

## 1. Introduction

Inflammatory bowel disease (IBD) is a chronic disorder of the lower gastrointestinal tract with unknown pathogenic factors, encompassing ulcerative colitis (UC) and Crohn’s disease (CD). Main symptoms of this chronic disease are diarrhea, weight loss, abdominal pain, rectal bleeding, and vomiting [1,2]. Currently, IBD has emerged as a global public health challenge, with accelerating incidence in developing areas such as Asia and South America. Although the incidence of IBD has stabilized in western countries, the burden remains significant as prevalence has exceeded 0.3%. IBD may result in hyperplasia and neoplastic growth [3]. The pharmaceutical treatments for IBD, including aminosalicylic acids, corticosteroids, immunomodulators, and biologic agents, have been widely used in clinical trials, and their efficacy, safety, and side-effects have also been reported quite extensively [4]. For instance, sulfasalazine (SASP) [5] is a sulfa antimicrobial that has been used to treat IBD in clinics for years, though it may cause nausea, anorexia, hemolysis, headaches, and other side effects. New treatments for IBD are urgently needed.

A variety of factors, such as environment [6], diet, genes [7], immunomodulatory [8], and gut microbiota [9], are known to be involved in the pathogenesis of IBD. Emerging evidence has proved that gut microbial dysbiosis is a major feature in IBD patients and mice colitis models [10], although involvement of specific bacterial species and the changes in the gut microbial ecology are under debate. It was reported that the occurrence of IBD is often associated with lower diversity of bacteria (*Faecalibacterium prausnitzii* [11,12], *Clostridium clusters IV* and *XIVa* [13,14], *Bifidobacterium* [15], *Bacteroides* [16], *Helicobacter pylori* [17], *Roseburia species* [18], and *Eubacterium restale* [19,20]) and decreased abundance of bacteria (*Escherichia coli* [21], *Fusobacterium* [22,23], and *Candida albicans* [24]), but a clear causal link of cause and effect has not yet been proven [25]. Studies have shown that traditional Chinese medicine (TCM) can regulate the gut microbiota during IBD treatment. It was reported that *Codonopsis pilosula* Nannf polysaccharides could increase three probiotics, namely *Bifidobacterium* spp., *Lactobacillus* spp., and *Akkermansia* spp., and decrease pathogenic bacteria, including *Desulfovibrio* spp., *Alistipes* spp., and *Helicobacter* spp. in mice with colitis [26]. Huangqin decoction increased the amount of *Lactobacillus* and decreased the abundance of *Desulfovibrio* and *Helicobacter* in the treatment of dextran sulfate sodium (DSS)-induced inflammation [27]. Considering the pivotal role of gut microbiomes in IBD, studying the altered microbe and microbial metabolism pathways linked to the host in the process of IBD treatment may help us further understand therapeutic efficacy and side effects.

Indigo naturalis (IN) (also known as Qing Dai), a blue powder that contains multiple ingredients, including indirubin, indigo, and tryptanthrin, is derived from the leaves and stems of indigo plants, such as *Baphicacanthus cusia* (Nees) Bremek. (Acanthaceae), *Polygonum tinctorium* Ait. (Polygonaceae), and *Isatis indigotica* Fort. (Cruciferae), and has been used to treat colitis in China for thousands of years [28]. The significant therapeutic effects of oral IN on IBD patients were also confirmed in a retrospective observational study [29,30], although pulmonary arterial hypertension (PAH) was observed in patients who received IN over 6 months. The patients recovered from the IN-induced PAH when they discontinued using IN. Recently, Makoto Naganuma et al. [31] found that IN may be useful in UC patients with steroid-dependent disease or previous use of anti-TNF-α. In addition, Chinese herbal medicines containing IN, such as Xilei-San [32,33], were also reported to be effective for treating UC in randomized, placebo-controlled trials. The efficacy of IN for treating colitis was also verified by means of regulating pro‑inflammatory factors and ameliorating colonic mucosal damage [34,35]. It was found that IN mainly consists of ligands of the aryl hydrocarbon receptors (AhR), such as indigo and indirubin [36,37]. Shoichiro Kawai et al. demonstrated that IN and indigo ameliorated murine colitis through AhR signaling activation [38]. Intriguingly, AhR signaling modulates the composition of the mucosal microbiota [39]. Our previous work has investigated the protective effects of tryptanthrin on colitis, which was closely related to TNF-α/NF-κB and IL-6/STAT3 pathways [40]. NF-κB and STAT3 play significant roles in the pathogenesis of IBD [41,42]. It was also reported that IN suppressed LPS-induced production of TNF-α and IL-6 in vitro [34].

Although IN exhibited significant clinical and endoscopic efficacy in treating colitis without causing serious adverse side effects [43,44], the concrete mechanisms of IN remain elusive [28]. The poor water solubility of IN greatly affects the absorption of its oral treatment in colitis. In addition, the reported evidence showed that IN could regulate the balance of *Bacteroides* and *Bifidobacterium* [29]. It is predicted that the protective effects of IN on colitis may be exerted through modifying gut microbiota. In order to evaluate how intestinal microbiomes contributes to IBD improvement with oral IN treatment, sequencing of 16S rDNA with the Illumina Microbiome Sequencing (MiSeq) platform was utilized and metagenomes of gut microbiota were inferred by the software package of Phylogenetic Investigation of Communities by Reconstruction of Unobserved States PICRUSt [45] to investigate differences in microbiome composition and function in the feces of DSS-induced mice with IN treatment, SASP was used as positive control because of its wide clinical applications.

## 2. Results

### 2.1. Identification of Chemical Components of IN

Under the optimized conditions described in Section 4.1, the major organic components in IN were well separated and detected within 25 min by ultra-performance liquid chromatography (UPLC). As shown in Figure 1, indigo and indirubin were analyzed, showing a satisfying degree of separation. The standard curves and linear ranges were as follows: indigo: *y* = 7 × 10^6^
*x* − 25751(*r* = 0.9999, 40.4~202.0 μg/mL); indirubin: *y* = 8 × 10^7^
*x* − 4734.7 (*r* = 0.9997, 3.552~35.520 μg/mL). Based on the external standard method, the amounts of indigo and indirubin were calculated and the results were 13.87% and 0.15%, respectively.

### 2.2. In Improved the Health Condition of Experimental Animals Suffering from DSS

Clinical activity score (CAS) was employed in our work to estimate the health status of the mice enrolled in our experiment. In addition, the CAS of each group (six samples for each group) reflected the DSS mouse model, along with the therapeutic effects of SASP (positive control) and three doses of IN. The results of Figure 2 showed the CAS of each group over 14 days. The line chart for the colitis group indicated that the colitis model treated with 3% DSS for 7 days was successful. After treatment with SASP and IN for 7 days, the CAS significantly decreased for both the SASP group and the three groups with different concentrations of IN, especially at the dose of 200 mg/kg, implying therapeutic effects of SASP and IN.

### 2.3. In Decreased the Levels of Pro-Inflammatory Cytokines and Increased Anti-Inflammatory Cytokines

Inflammatory cytokines play important roles in the occurrence and progression of colitis. The balance of pro-inflammatory and anti-inflammatory cytokines is essential for maintaining a healthy condition in humans. In our experiment, four inflammatory cytokines were detected by Elisa kit, with six samples taken for each group. The results are shown in Figure 3. Levels of pro-inflammatory cytokines IL-6, IL-8, and TNF-α were all increased extremely significantly in colitis group compared with the negative control (CON). Conversely, the level of anti-inflammatory cytokine IL-10 decreased significantly (comparing with the control group) in the colitis group. However, after treatment with SASP and IN, the levels of IL-6, IL-8, and TNF-α decreased greatly compared with the colitis group, while the levels of IL-10 increased. It can be seen that the therapeutic efficacy with IN at the dose of 200 mg/kg was even better than that with SASP. Therefore, the medium dose group was adopted for follow-up experiments and analysis.

### 2.4. In Improved the Morphological Structure of Colitis

The results in Figure 4 using hematoxylin and eosin (H&E) staining exhibited the histologic characteristics of each group in our experiment. The intact colon structure, including mucosa, submucosa, inner circular muscle, and outer longitudinal muscle, was clearly observed, with the crypt and goblet cells regularly arranged in the control group, as shown in Figure 4A. Figure 4B shows that 3% DSS in drinking water caused extensive colonic tissue damage, including the distortion and atrophy of the crypt, epithelial cell necrosis and edema, neutrophil infiltration, and lymphocyte accumulation between the basement of the crypt and the mucous muscle. The results in Figure 4C show that treatment with SASP at the concentration of 125 mg/kg ameliorated the colitis condition induced by DSS. However, crypt branching and atrophy still existed and many inflammatory cells were still observed in mucosa, implying the damage to colon tissues caused by inflammatory cell infiltration was not completely repaired. Figure 4D illustrates the treatment effects of IN at the concentration of 200 mg/kg. The neutrophil infiltration, crypt branching, distortion, and atrophy were also present. The columns in Figure 4E show the histopathological scores of each group. The column heights of SASP and IN are considerably lower than that of the colitis group, indicating the good therapeutic effect.

### 2.5. In Ameliorated the Intestine Microecology of Colitis Mice Induced by DSS

To determine the effects of IN on the gut microbiota, 16S rDNA gene amplification (V4-V5 region) and sequencing were employed with the Illumina MiSeq platform in our work (see Materials and Methods, Section 4.7). Species accumulation curves were used to compare diversity properties of community data sets using different accumulation functions. The specaccum function can find species accumulation curves or the numbers of species for a certain number of sampled sites or individuals. As shown in Figure 5A, specaccum species accumulation curves gradually increased as sequenced samples increased and finally became saturated, indicating that 12 samples (*n* = 3 for each group) were enough to include the majority of gut microbiota in mice. Rank–abundance curves can be used to visualize species richness and species evenness. The results shown in Figure 5B also confirmed that the sequencing depth was suitable for our experiment. A total of 928 operational taxonomic units (OTUs) were generated from 12 samples, representing 928 ecological groups, which were then identified to 20 phyla and 159 genera. According to the petal pattern of OTU samples (Figure 5C), 189 OTUs coexisted among 12 samples, and more than 200 OTUs were present in each sample. The Venn diagram displayed in Figure 5D reveals that 429 OTUs overlapped among four groups, which was far more than the number of bacteria in each group, indicating that the analysis of the gut microbiota in the four groups could obtain more reliable results.

Alpha and beta diversity analyses was conducted using Quantitative Insights Into Microbial Ecology (QIIME). Alpha diversity was calculated by different indices of species richness and evenness, such as Chaol (an estimator of species richness based on a vector or matrix of abundance data), observed_species, PD_whole_tree (Faith’s phylogenetic diversity), and Shannon indices. The results shown in Figure 6A illustrate that the abundance and diversity of gut microbiota in the model group were all increased. After treatment with IN, reverse trends were observed, indicating that the protective effects of IN were closely related to the abundance and diversity of gut microbiota. The results of the Shannon index calculations indicated that the community diversity decreased in both the model and SASP groups. Beta diversity is a term for the comparison of samples with each other. Principal component analysis (PCA) (Figure 6B) revealed that microbiota communities among control, DSS, and IN groups were clearly separated, but were not separated between positive and model groups. The system clustering tree (Figure 6E) showed that the level of the model group was close to that of the positive group, except for P1 (one of the positive samples). There was a significant difference between the IN group and the others. To some extent, the control group and other treated groups were able to cluster into different classes, indicating that DSS, SASP, or IN treatment had a specific impact on the intestinal flora of mice. We further investigated the relative abundance of the main phyla (Figure 6C) and genus (Figure 6D) in each group. *Bacteroidetes, Firmicutes, Proteobacteria*, and *Deferribacteres* were the dominant phyla found in all groups. Increased *Bacteroidetes* and *Proteobacteria* and decreased *Firmicutes* were observed in the DSS-induced colitis model compared with the control group, which was in accordance with the rising *Bacteroidetes/Firmicutes* ratio of colitis in the previous report [46,47,48]. The results demonstrated that *Bacteroidetes* and *Proteobacteria* were positively correlated with TNF-α, IL-6, and IL-8, which promoted the occurrence of IBD, while *Firmicutes* was correlated with IL-10, which inhibited IBD. However, after IN treatment, the abnormal microbiota community structure of colitis mice was similar to the control group. At the genus level, the most dominant bacterium in all groups were *Prevotellaceae_UCG_001, uncultured_bacterium, Alloprevotella, Bacteroides, Lachnospiraceae_NK4A136_group, Allobaculum, Alistipes, Phascolarctobacterium, Lactobacillus*, and *Dubosiella*, which accounted for about 80% of the abundance in total (Figure 6D).

LDA effect size (LefSe) analysis was used to detect biomarkers and dominant microbiota in each group, with an effect size threshold of 2. The results shown in Figure 7 revealed that in DSS group, 8 genera were markedly higher and 13 genera were markedly lower than those in the control group (Figure 7A). Compared with DSS group, 8 genera were greatly higher and 7 genera were lower in the IN group (Figure 7B). Altogether, the abundance of *Turicibacter* was increased, while the abundances of *Peptococcus, Gemella*, and *Family_XIII_UCG_001* genera were reduced in colitis mice and adjusted to the control level after treatment with IN (Figure 7D). However, the relative abundances of *Gemella* and *Family_XIII_UCG_001* genera changed slightly. Above all, *Turicibacter* and *Peptococcus* may be closely related to the role of IN in alleviating colitis. Compared with DSS group, 5 genera were greatly higher and 9 genera were lower in the SASP group (Figure 7C). The relative abundance of *Staphylococcus*, *Jeotgalicoccus*, and *Eubacterium_nodatum_group* were reduced in colitis mice, while they were adjusted to the control level with SASP, as shown in Figure 7D. However, treatment with both IN and SASP up-regulated the relative abundance of *Eubacterium_nodatum_group*, alleviating colitis. In summary, it can be seen that there was a great difference between IN and SASP in regulating the structure of gut microbiota in DSS-induced colitis due to their different effective mechanisms.

### 2.6. In Treatment Changed the Metabolic Genes in the Metagenomes

Metagenomes of gut microbiota are the basis of their metabolic functions. Since different treatment significantly changed the microbial community, we then investigated gut microbiota functions. Phylogenetic Investigation of Communities by Reconstruction of Unobserved States (PICRUSt) software was used in our work and the predicted metagenome information was then collapsed into the KEGG pathway (level 3) based on the 16S rDNA sequences. STAMP package was used for functional profiling. In total, 9 pathways were changed in colitis mice (Figure 8A). The gene abundances in pathways of phenylalanine metabolism, other transporters, ascorbate and aldarate metabolism, lysine degradation, arachidonic acid metabolism, biosynthesis of siderophore group nonribosomal peptides, non-homologous end-joining, and glycosphingolipid biosynthesis of lacto and neolacto series were increased, while only the limonene pathway and pinene degradation were deceased. Compared with the model group, only the insulin signaling pathway was decreased after SASP treatment (Figure 8B), while 8 pathways were changed after IN treatment (Figure 8C). In total, 4 pathways were up-regulated, including amino sugar and nucleotide sugar metabolism, fructose and mannose metabolism, nucleotide metabolism, and polycyclic aromatic hydrocarbon degradation, while 4 pathways were down-regulated, including the vibrio cholera pathogenic cycle, biosynthesis of siderophore group nonribosomal peptides, non-homologous end-joining, and glycosphingolipid biosynthesis of lacto and neolacto series. Therefore, it was predicted that the protective effects of IN on colitis were closely related with the biosynthesis of siderophore group nonribosomal peptides, non-homologous end-joining, and glycosphingolipid biosynthesis of lacto and neolacto series.

## 3. Discussion

IBD is a severe health problem and negatively affects the quality of human life. With the exception of corticosteroids, conventional treatment with active UC in patients is quite expensive. Chinese herbals in combination with IN have been used to treat UC patients and to remedy inflammatory conditions. It was reported that inorganic components accounting for 90% in IN are mainly CaCO_3_, SiO_2_, and H_2_O, among others. In this study, the organic chemical compositions of IN were detected by UPLC and the major components of indigo and indirubin were well detected. It was clearly proven in other studies that oral treatment with indigo and indirubin could improve murine colitis [37,38]. The results of the CAS suggested that IN treatment with concentrations ranging 100–400 mg/kg could greatly improve the health conditions of the colitis mice.

Until now, although the occurrence and development of IBD has been demonstrated to be related to various factors, the imbalance of inflammatory cytokines plays a significant role for the entire duration of IBD. To evaluate the intestinal immunity regulation effects, four inflammatory cytokines, namely IL-6, IL-8, TNF-α, and IL-10, were investigated. The results showed that the treatment with IN could significantly down-regulate pro-inflammatory cytokines and up-regulate anti-inflammatory cytokines. H&E staining was used to confirm the colon damage from DSS-induced colitis and the therapeutic effects of SASP and IN. It was shown that IN could greatly improve the histopathological structure of colitis induced by DSS. The 200 mg/kg treatment of IN was even better than that with SASP.

Colitis is mainly characterized by tissue damage, colon inflammation, and imbalance of gut microbiota. The gut microbiota have a great impact on disease phenotypes and activity in mice colitis models. Gut microbiota-based therapeutic approaches have also shown effects in amelioration of colitis models. Our results showed that IN could greatly reduce the alpha diversity of the gut microbiota community compared with DSS group, indicating that the therapeutic effects of IN on colitis were closely related to modulating gut microbiota. Moreover, the PCA and system clustering tree results showed significant distances between each group except SASP, which was consistent with the previous study showing that DSS treatment could change the beta diversity of the gut microbiota community [27]. The gut microbiota community in all samples was evaluated based on the following criteria: phylum, class, order, family, and genus. The rising *Bacteroidetes*/*Firmicutes* ratio of colitis was decreased after IN treatment (200 mg/kg), suggesting that IN could treat the DSS-induced gut microbiota disruption.

The results of LefSe analysis showed that IN could decrease the relative abundance of *Turicibacter* while increasing *Peptococcus* in the colitis group. The *Turicibacter* genus is a strictly anaerobic and gram-positive bacteria that produces lactic acid by fermenting sugars, and is commonly detected in the gastrointestinal tracts and feces of humans and animals. It has been proven that *Turicibacter* positively correlated with pro-inflammatory cytokines, such as IL-6, IL-1β, TNF-α, and IFN-γ [49,50]. *Peptococcus* is also an anaerobic gram-positive coccal bacteria in the family *Peptococcaceae*, which is part of the normal flora of the mouth, upper respiratory tract, and large intestine. *Peptococcus* does not ferment carbohydrates but does so for proteins, peptides, and amino acids. It can be isolated in all types of anaerobic infections and is generally recovered mixed with other aerobic or anaerobic organisms [51]. It also can be seen from the LefSe results that the up-regulation of the relative abundance of *Staphylococcus*, *Jeotgalicoccus*, and *Eubacterium_nodatum_group* in colitis mice may be related to the therapeutic effect of SASP. SASP is a prodrug that is split in the colon by bacterial azoreductases to release 5-aminosalicylic acid (5-ASA) and sulfapyridine, which has been found to treat 2,4,6-trinitrobenzenesulfonic acid (TNBS)-induced gut dysbiosis [52]. The inferred metagenomes from 16S data using PICRUSt demonstrated the decrease of metabolic gene pathways, such as biosynthesis of siderophore group nonribosomal peptides, non-homologous end-joining, and glycosphingolipid biosynthesis of lacto and neolacto series, may allow the microbiota to maintain homeostasis during inflammation from IN treatment for DSS-induced colitis.

In summary, our study demonstrated that IN could significantly alleviate the severity of inflammation, modulate the dysregulated metabolism pathways in mice with colitis, and restore the unbalanced microbiota composition of colitis mice to a normal condition. The therapeutic effect of IN may be closely related with the anaerobic gram-positive bacteria of *Turicibacter* and *Peptococcus.* Our data provide a new insight into gut microbiota activity to better understand the therapeutic efficacy of IN for ameliorating colonic inflammation. However, the inferred metagenomes based on the 16S data ought to be validated by shotgun metagenomic sequencing. The limitations of the inferred metagenomes should also be considered. The current 16S sequencing mainly includes bacterial and archaeal genomes, ignoring non-bacterial organisms, such as viruses, fungi, and phages. To further explore the relevant genes in microbial communities, shotgun metagenomic sequencing, proteomics, and metabolomics should be taken into consideration. Moreover, more samples are necessary to explore the heterogeneity of microbial communities within individuals.

## 4. Materials and Methods

### 4.1. UPLC Analysis

IN granules were purchased from Jiangyin Tian Jiang Pharmaceutical Co., Ltd. The chemical components of IN were identified by UPLC. Each sample was weighed and ultrasonically extracted (300W, 40 kHz, 30 °C) with N, N-Dimethylformamide (DMF) for 60 min in a KQ-300DE CNC ultrasonic cleaner bath (Kunshan Ultrasound Instrument Co., Ltd., Jiangsu, China). The extraction was filtered through a 0.22 μm microporous filter for subsequent UPLC analysis.

Chromatographic separation was performed using a Waters Acquity UPLC H-Class system (Milford, MA, USA) consisting of a quaternary gradient ultra-high pressure infusion pump, a standard auto-sampler, a column oven, and photodiode array detectors (DAD). An XBridge BEH C18 column (2.1 mm × 150 mm, 2.5 μm, Waters Corp., Milford, MA, USA) was employed in this experiment. The mobile phase consisted of (A) 0.1% formic acid in water and (B) acetonitrile. A linear gradient was optimized as follows (flow rate, 0.35 mL/min): 0–1 min, 5% B; 1–3 min, 5–40% B; 3–5 min, 40–60% B; 5–8 min, 60–62% B; 8–11 min, 62–65% B; 11–13 min, 65–70% B; 13–15 min, 70–95% B; 15–18min, 95% B; 18–20 min, 95–5% B; 20–25 min, 5% B. The injection volume was 5 μL and the monitoring was performed with an UV wavelength of 285 nm at 35 °C.

### 4.2. DSS Mouse Models

Male Kunming (KM) mice (SCXK (Sichuan) 2015-030) weighing 20 ± 2 g were purchased from Laboratory Animal Center of Xi’an Jiao Tong University Health Science Center. The animal studies were conducted according to protocols approved by Institutional Ethical Committee of Shaanxi University of Chinese Medicine. Animals were maintained under standard laboratory conditions and were given autoclaved ultra-filtered water and animal feed ad libitum. Animal handling and scoring of colitis were performed in a blinded experimental design.

DSS (40,000 Da) was purchased from MP biomedicals (Irvine, CA, USA) and dissolved in distilled water. After one week of adaptation feeding, mice were randomly divided into different groups (six mice for each group). Colitis was induced by providing drinking water containing 3% DSS for 5 to 7 days. Control mice received distilled water. Positive control mice were provided SASP at a concentration of 125 mg/kg through intragastric administration for 7 days [53]. IN was administered orally twice a day at high (100 mg/kg), medium (200 mg/kg), and low (400 mg/kg) concentrations for 7 days.

### 4.3. Clinical Activity Score

CAS involving animals’ body weight loss, stool consistency, and rectal bleeding was recorded daily during the experiment, as described previously [54]. The total clinical score ranged from 0.0 (healthy) to 4.0 (maximal activity of colitis).

### 4.4. Cytokine Assays

The experimental animals were sacrificed after 14 days and eyeballs were excised to take blood. Subsequently, the blood was centrifuged to test the levels of IL-6, IL-10, IL-8, and TNF-α using Elisa kits according to the manufacturer’s instructions. The Elisa kits for IL-6, IL-8, IL-10, and TNF-α were obtained from Neobioscience and the batch codes were M171109-604a, M171109-104a, M171109-605a, and M171109-102a, respectively.

### 4.5. Histopathological Evaluation

The colon was spread onto a plastic sheet, fixed with 10% neutral-buffered formalin for 48 h, and embedded in a paraffin block. Sections of paraffin-embedded tissues were subjected to hematoxylin and eosin (H&E) staining to measure the severity of inflammation. Histologic scoring was performed by a pathologist. For infiltration of inflammatory cells, rare inflammatory cells in the lamina propria were counted as 0; increased numbers of inflammatory cells in the lamina propria as 1; confluence of inflammatory cells extending into the submucosa as 2; and a score of 3 was given for transmural extension of the infiltrate. For tissue damage, no mucosal damage was counted as 0, discrete lymphoepithelial lesions were counted as 1, surface mucosal erosion was counted as 2, and a score of 3 was given for extensive mucosal damage and extension through deeper structures of the bowel wall. The combined histologic score ranged from 0 (no changes) to 6 (extensive cell infiltration and tissue damage).

### 4.6. DNA Extraction

The control, model, positive, and IN groups (middle dose) were selected for gut microbiota study, with three samples taken for each group. The intestinal contents of each mice were collected in sterilized, airtight tubes on ice before storing at −80 °C for further analysis. Genomic DNA was extracted from intestinal content samples using QIAamp DNA Stool Mini kit (Qiagen, Germany). DNA integrity and quantity were assessed with agarose gel electrophoresis (concentration of agarose gel: 1%; voltage: 170 V; electrophoresis time: 30 min) and a nanodrop instrument (Thermo Fisher Scientific), respectively. From each sample, 30 ng purified DNA was used for the subsequent polymerase chain reaction (PCR) and 16 S rDNA sequencing.

### 4.7. 16S rDNA Amplification and Sequencing

The 16S rDNA amplification of the V4–V5 (515–909) region was conducted by PCR (ABI 9700, USA), using two universal bacteria 16S rDNA primers: forward GTGCCAGCMGCCGCGGTAA and reverse CCCCGYCAATTCMTTTRAGT [55]. PCR products were purified with agarose gel electrophoresis and quantified with QuantiFluor™-ST (Promega, Madison, WI, USA). The gene library was constructed using NEBNext Ultra II DNA Library Prep Kit (NEB, Ipswich, MA, USA). Sequencing was performed using an Illumina MiSeq system (Illumina, San Diego, CA, USA), according to the manufacturer’s instructions.

Raw data from sequencing was subjected to FLASH software (version 1.2.11, CBCB, Baltimore, MD, USA) for paired-end read assembly [56], then quality filtered with Trimmomatic (version 0.35, Aachen, Germany) [57] and USEARCH (version 8.1.1861, Tiburon, CA, USA) [58] according to literature. Clean tags were clustered with UPARSE [59] at 97% similarity level and 0.005% threshold value, at which point OTUs were finally generated. The OTUs were then identified as different taxonomies with the Ribosomal Database Project (RDP) Classifier (version 14, East Lansing, MI, USA) [60] at the confidence threshold of 70%. The 16S ribosomal RNA database was downloaded from the Silva website (http://www.arbsilva.de, Release 128) [61]. The abundance of bacteria in OTUs was calculated as the proportion of their clean tags out of the number of total clean tags.

### 4.8. Diversity Analysis, Differential Analysis, and Co-Occurrence Network Analysis

Alpha diversity of each sample, including calculations with Chaol, observed_species, PD_whole_tree, and Shannon indices, were transformed from OTUs using QIIME (version 1.8, Flagstaff, AZ, USA) [62]. The alpha significant value for non-parametric tests among groups was set at 0.05 to give more accurate results. For beta diversity analysis, an OTU table of all samples was computed with beta diversity metrics in order to conduct PCA [63]. We employed the online tool LefSe to explore significantly affected bacteria based on genus abundance [64]. The threshold for the logarithmic liner discriminant analysis (LDA) score was set at 2, and other parameters were set as default.

### 4.9. Metagenomes Prediction and Analysis

PICRUSt was designed to predict metagenome functional content from 16S rRNA genes [45]. According to the literature, the predicted metagenomes are highly consistent with sequenced metagenomes. Therefore, we employed this software to calculate metagenomes of each sample, strictly according to the manual instructions. Then, the metagenomes were collapsed into the level 3 KEGG pathway and analyzed using the STAMP package (version 2.1.3, Halifax, Nova Scotia, Canada) [65].

### 4.10. Statistical Analysis

Each experiment was repeated at least three times, and the data are presented as the mean ± SD. Statistical differences between groups were examined using SPSS 19.0 program with Student’s t-test or the Mann–Whitney U test as appropriate. Here, *p* values less than 0.05 were recognized as significant and values less than 0.01 were recognized as very significant for the genus abundance and metagenome data. Bar charts and scatterplots were plotted in GraphPad Prism 6 software. Venn diagrams were generated using an online tool (http://bioinfogp.cnb.csic.es/tools/venny). The extended error bar plot was generated in STAMP software.

## Figures and Tables

**Figure 1 molecules-24-04086-f001:**
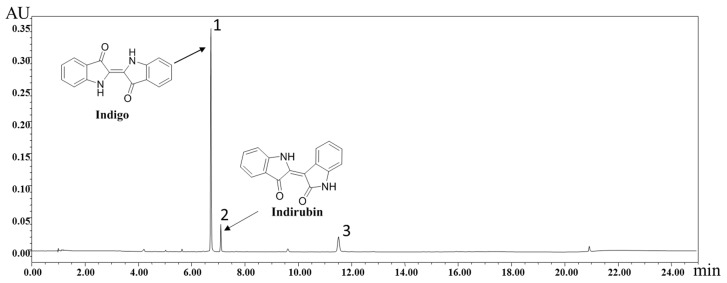
Ultra-performance liquid chromatography (UPLC) chromatograms of indigo naturalis (IN) granules: peaks 1–3 are derived from indigo (6.79 min), indirubin (7.16 min), and column (11.58 min), respectively.

**Figure 2 molecules-24-04086-f002:**
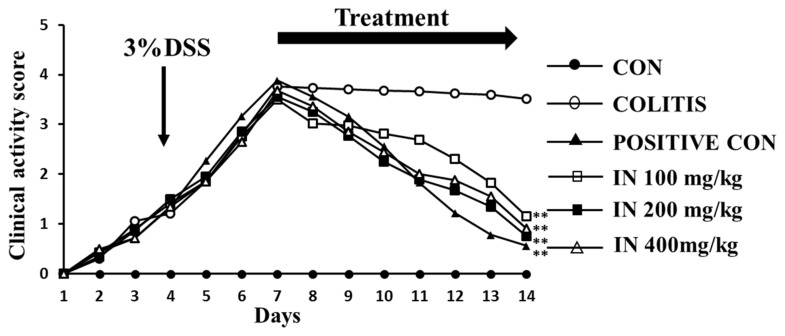
Clinical activity score (CAS) of mice. After colitis induction, the animals were treated for 7 days with distilled water (CON as “control”), 3% dextran sulfate sodium (DSS, as COLITIS), 125 mg/kg sulfasalazine (SASP, as “POSITIVE CON”) and three concentrations (100 mg/kg, 200 mg/kg, and 400 mg/kg) of IN, respectively. The untreated group showed no improvement after 14 days, while treatment with SASP and three concentrations of IN significantly improved the CAS of colitis mice. Note: *n* = 6; ** *p* < 0.01 compared with colitis.

**Figure 3 molecules-24-04086-f003:**
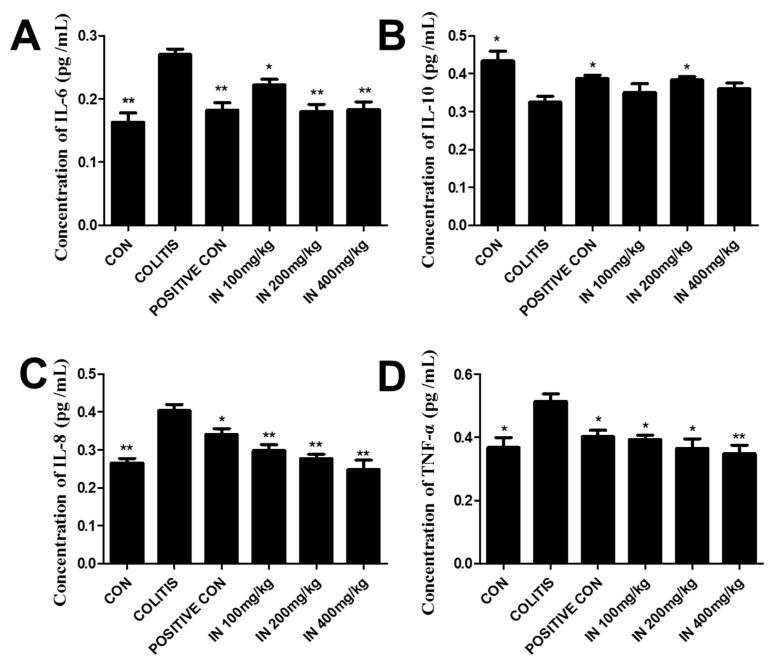
Levels of pro-inflammatory cytokines IL-6, IL-8, and TNF-α and anti-inflammatory cytokine IL-10 in each group. (**A**) The concentration of IL-6 in each group. (**B**) The concentration of anti-inflammatory cytokine IL-10 in each group. (**C**) The concentration of IL-8 in each group. (**D**) The concentration of TNF-α in each group. After treatment with SASP and three concentrations of IN, the levels of IL-6, IL-8, and TNF-α decreased sharply, and the levels of IL-10 increased inordinately. Error bars represent means ± SD; *n* = 6; * *p* < 0.05 and ** *p* < 0.01.

**Figure 4 molecules-24-04086-f004:**
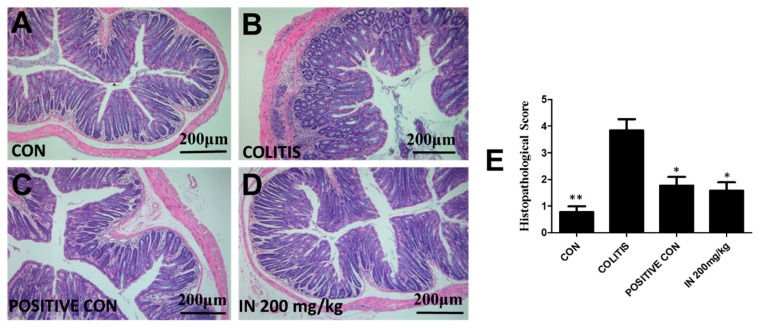
Histologic characteristics of each group (magnification ×100). (**A**) Normal colonic mucosa, with regularly formed colonic folds covered by intact mucosa. (**B**) The histologic characteristics of colitis group. Inflammatory cells completely infiltrated the mucosa and submucosa. (**C**) The histologic characteristics of SASP group. (**D**) The histologic characteristics of IN group at the concentration of 200 mg/kg. (**E**) The sum of the histological scores based on blinded histopathological analysis. Error bars represented means ± SEM; *n* = 6, * *p* < 0.05 and ** *p* < 0.01.

**Figure 5 molecules-24-04086-f005:**
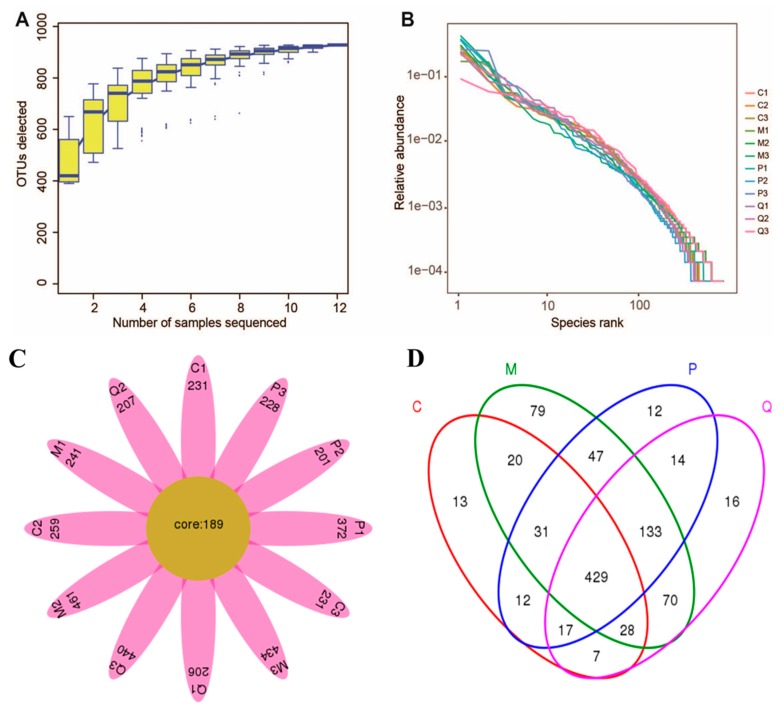
Evaluation of Illumina MiSeq sequencing data, showing that 12 samples and the selected sequencing depth were sufficient to obtain reliable results. (**A**) Specaccum species accumulation curves determined at the 97% similarity level. (**B**) Rank-Abundance curves of the 12 samples. (**C**) Petal pattern of operational taxonomic units (OTUs). (**D**) Venn diagram indicating different numbers of OTUs in the four groups. Note: C = control group; M = DSS model group; P = SASP group at 125 mg/kg); Q = IN group at 200 mg/kg, Here, *n* = 3 for each group.

**Figure 6 molecules-24-04086-f006:**
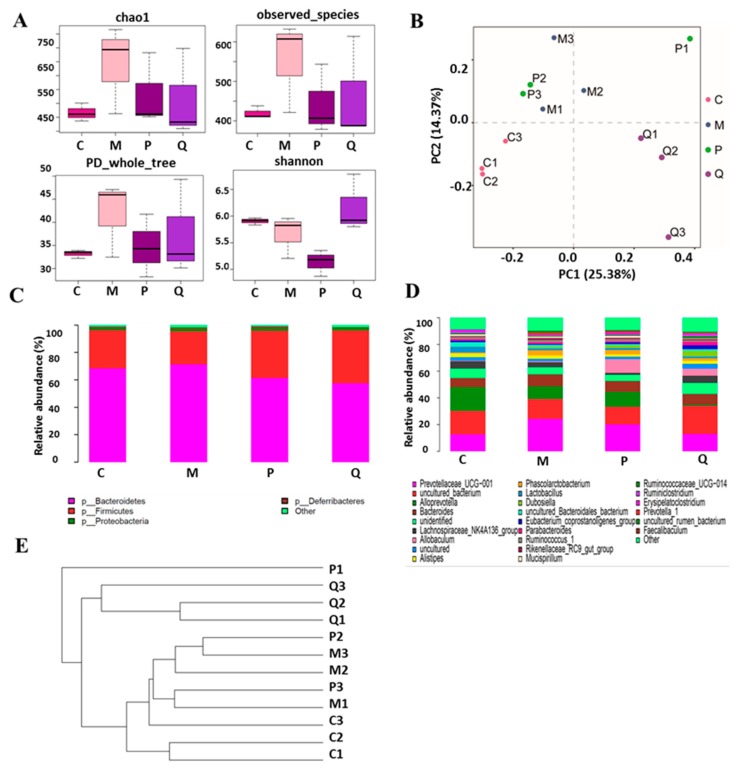
IN treatment changed the structure of gut microbiota in DSS model mice. (**A**) Alpha diversity index estimated the richness and diversity of colon contents and bacterial populations. (**B**) Multiple sample principal component analysis (PCA). (**C**) Relative abundance of the main phylum in the intestinal microbiota. (**D**) Relative abundance of the main genus in the intestinal microbiota. (**E**) System clustering tree of gut microbiota based on unweighted Unifrac metrics indicating the beta diversity of gut microbiota in each group. Note: C = control group; M = DSS model group; P = SASP group at 125 mg/kg; Q = IN group at 200 mg/kg. Here, *n* = 3 for each group.

**Figure 7 molecules-24-04086-f007:**
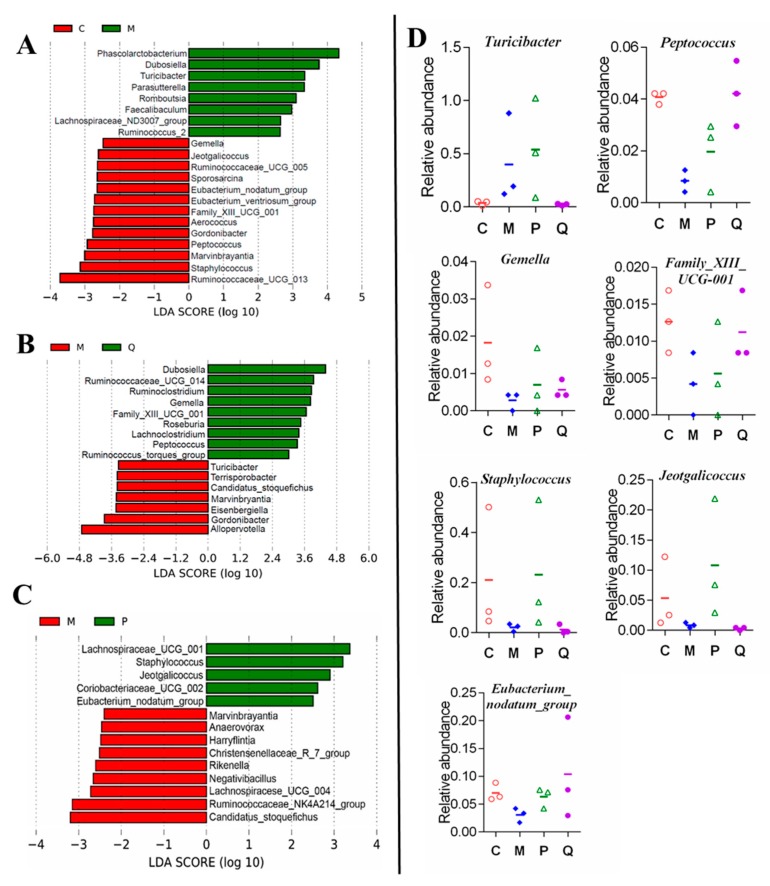
Difference in dominant microorganisms between groups in LDA effect size (LefSe) analysis based on genus abundance (*n* = 3), showing the genera with LDA scores of more than 2 and a significant value less than 0.01. (**A**) Control and model groups. (**B**) Model and IN groups. (**C**) Model and positive groups. (**D**) Abundance of genera that is significantly changed by IN and SASP treatment. The percentages were calculated from the number of sequenced tags for each genus out of the total tags. Note: C = control group; M = DSS model group; P = SASP group; Q = IN group.

**Figure 8 molecules-24-04086-f008:**
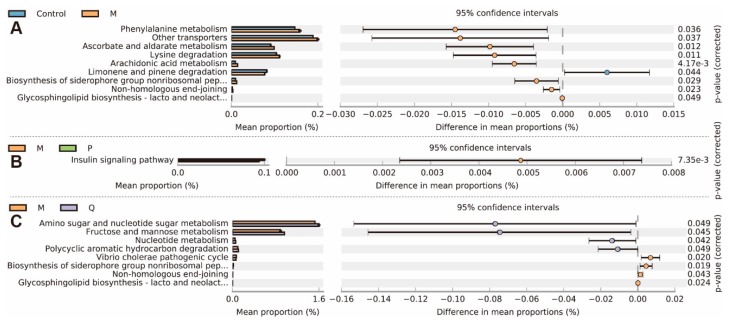
The changes in the metabolic genes in the metagenomes induced by different treatments. (**A**) Control and model (M) groups. (**B**) Model (M) and positive (P) groups. (**C**) Model (M) and IN (Q) groups. Extended error bar plot of genes (categorized by level 3 KEGG pathways) was made using STAMP software, while significant values between groups were calculated by Wilcoxon test, with 0.01 set as the significance threshold (*n* = 3).

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
