# Peer review of "Indigo Naturalis Ameliorates Dextran Sulfate Sodium-Induced Colitis in Mice by Modulating the Intestinal Microbiota Community"

_molecules, 2019, doi:10.3390/molecules24224086_

Round 1

Reviewer 1 Report

Overall, the idea of the paper is novel and pertinent to the current literature.

The manuscript might benefit of another proofread for grammar, structure and style.

The abstract should be more concise, by clearly mentioning the purpose of the work, the materials and methods used and the main findings which are not very well highlighted. The introduction is quite well organized, but it also needs some important modifications. Please pay attention at the spaces between words for all the article! Focus more on the current state of the art regarding the effects of indigo naturalis on colitis by highlighting the results of the studies. Also it is important to mention which are the active substances from IN which exhibit the protective effects but at the same time mention also the controversial/diverging results of studies focusing on the adverse effects of indigo naturalis in colitis.

The experimental results are well organized, however the differences between the treatment with sulfasalazine and IN are not very well explained, also the sample size is not clearly mentioned. Also which were the treatment effects of the other doses of (100mg/kg and 400 mg/kg) of IN on the morphological structure of colitis?

The discussion section also needs modifications. Is it not very well highlighted how your results are interpreted in perspective of previous studies and of the working hypotheses. The limitations of the work also need to be highlighted. Future research directions may also be mentioned.

Materials and methods is well structured.Please also mention the sample size taken into consideration at the DSS mouse models.

Specific comments:

Title: please also mention in the title that the study reports animal trial data

Line 19: „The chemical compositions of IN were detected...” instead of was.

Line 20: „the protective effects on colitis were determined”

Line 34, 35: what is the prevalence, incidence?

Line 42: Please reformulate „the occurrence of IBD is often associated with decreasing   potential   beneficial microbiota...” -Lower diversity of the bacteria/decreased abundance

Reference 12 does not state that those bacterial species are potential beneficial.

Line 49: Please reformulate „in colitis mice” with in mice with colitis.

Line 50: Italic style for Lactobacillus

Line 51: decrease abundance of Desulfovirbio...

Line 52: please mention from which part of the plan tis indigo naturalis extracted and if it is a formulation.

Line 53:please reformulate ” And its therapeutic effects of IN on colitis was” -the therapeutic effects ...were...

Name also the therapeutic effects mentioned in this study

Please correct with „thousands of years”.

Line 59: please specify why these pathways are important.

Line 63: italic font for the bacteria names.

Line 81: specify why you chose the treatment with sulfasalazine as positive control.You could also specify in the introduction that is is a a medication used to treat ulcerative colitis.

Line 129: please also specify what are the specaccum species

Line 130: were instead of was

Line 148: were instead of was

Line 200: decreased instead of deceased

Line 222: please provide references

Line 247: bacteria, without „s”

Author Response

Dear reviewer,

    Please see the attachment, thank you!

Sincerely yours

Dr. Yan-Ni Liang

18th, Oct. 2019

Shaanxi University of Chinese Medicine, Xianyang, 712083, China

e-mail: aiziji_2005@126.com

Information of the corresponding authors is as follows:

Zhi-Shu Tang, Shaanxi University of Chinese Medicine, Xianyang, 712083, China, e-mail: tzs6565@163.com, Tel: 86-29-38185060 Zheng Wang,Shaanxi University of Chinese Medicine, Xianyang, 712083, China, e-mail: wazh0405@126.com, Tel: 86-29-38182201

Reviewer 2 Report

Overall the manuscript is rather poorly presented.

Minor concern:

There are some spelling mistakes; e.g.:

extracted form – to - extracted from For thousands years – to - for thousands of years

All latin terms should be intalic.

The tense is fluctuating in the introduction.

The authors need to explain the relevance of the equations y=8×10^7 x-4734.7, r2=0.9994, 3.552~35.52;

I would question the relevance of the decimals that are given.

Similarly, peak retention time three decimals are not very meaningful

SASP need to be introduced in full

Fig2 need X-axe legend

CON needs to be explained (controls are normally abbreviated ctr)

The sentence on line 92 is incomplete.

C, M, P, Q should also be introduced in fig 5 legende, the font on the figure is also too small

Major concern:

The result section is a good example of how the results are summarized figure by figure. The authors literary writes; Fig x shows this ….. Fig Y shows that.. Instead the authors should try to motivate their studies, why are they interested in investigating this, how do they check it and then what did they find (Fig X). This will then hopefully lead into the next section, thorough a bridge, where a new argument is brought up, a new solution or metrology and then the authors findings (Fig Y). That style would make is much easier to read, and also easier to judge if the authors are using the right techniques. It will also make it harder for the authors not to fully present and interpret, thus not leaving it up to the reader to draw the conclusions.

The discussion is fairly underdeveloped.

Author Response

Dear reviewer,

   please see the attachment, thank you!

Sincerely yours

Dr. Yan-Ni Liang

18th, Oct. 2019

Shaanxi University of Chinese Medicine, Xianyang, 712083, China

e-mail: aiziji_2005@126.com

Information of the corresponding authors is as follows:

Zhi-Shu Tang, Shaanxi University of Chinese Medicine, Xianyang, 712083, China, e-mail: tzs6565@163.com, Tel: 86-29-38185060 Zheng Wang,Shaanxi University of Chinese Medicine, Xianyang, 712083, China, e-mail: wazh0405@126.com, Tel: 86-29-38182201

Round 2

Reviewer 1 Report

Changes accepted.